# Post-Herpetic Anti-NMDAR Encephalitis in Denmark: Current Status and Future Challenges

**DOI:** 10.3390/biomedicines12091953

**Published:** 2024-08-27

**Authors:** Anna Søgaard, Charlotte Aaberg Poulsen, Nadia Zeeberg Belhouche, Alberte Thybo, Siv Tonje Faret Hovet, Lykke Larsen, Christine Nilsson, Morten Blaabjerg, Mette Scheller Nissen

**Affiliations:** 1Department of Clinical Research, University of Southern Denmark, 5230 Odense, Denmark; annamagnussen1610@gmail.com (A.S.); tonjehovet@gmail.com (S.T.F.H.); lykke.larsen@rsyd.dk (L.L.); christine.nilsson@rsyd.dk (C.N.); morten.blaabjerg1@rsyd.dk (M.B.); 2Department of Nuclear Medicine, Odense University Hospital, 5000 Odense, Denmark; charlotte.aaberg.poulsen@rsyd.dk; 3Department of Infectious Medicine, Odense University Hospital, 5000 Odense, Denmark; 4Department of Clinical Immunology, Odense University Hospital, 5000 Odense, Denmark; 5Department of Neurology, Odense University Hospital, 5000 Odense, Denmark

**Keywords:** *N*-methyl-D-aspartate receptor encephalitis, post-viral autoimmune encephalitis, herpes simplex virus 1 Encephalitis, autoimmune encephalitis, viral encephalitis

## Abstract

It is well known that *N*-methyl-D-aspartate receptor encephalitis (NMDARE) can be triggered by infectious encephalitis such as herpes simplex virus 1 encephalitis (HSE). However, the incidence of post-HSE NMDARE in Denmark is unknown. We reviewed literature cases and compared these to retrospectively identified cases of post-HSE NMDARE in Denmark, using a national cohort database of autoimmune encephalitis (AE) and two regional databases of infectious encephalitis patients. We identified 80 post-HSE NMDARE cases in the literature, 66% being children, who more often presented movement disorders, decreased consciousness, and sleep disturbances compared to adults. Eight patients with post-HSE NMDARE were identified from the national cohort database of AE, none being children. Forty-four HSE patients were identified from the regional infectious encephalitis databases. Of these, 16 (36%) fulfilled the Graus criteria for probable/definite NMDARE, and eight (18%) presented a prolonged/relapsing disease course. Ten (23%) were tested for AE during hospitalization. Six (14%) had leftover cerebrospinal fluid available for retrospective autoantibody testing. One out of these six patients (17%) harbored NMDARE antibodies. Thus, in total, nine post-HSE NMDARE patients have been identified in Denmark from 2009 to 2021. Comparing the adult Danish patients to the literature, Danish patients were older, but the clinical phenotype and paraclinical findings were similar. Overall, the incidence of adult post-HSE NMDARE in the Region of Southern Denmark was 0.17 per million people per year and only 7% of adult HSE patients in the region were diagnosed with post-HSE NMDARE. Our findings suggest that adult patients are still underdiagnosed and the absence of pediatric cases diagnosed with post-HSE NMDARE in Denmark is highly concerning.

## 1. Introduction

Encephalitis is an acute severe inflammatory brain disease, affecting patients of all ages, with an incidence of six cases per 100,000 people per year in Western countries [1]. There are many possible etiologies, with herpes simplex virus 1 (HSV) being the most frequent [2,3]. In a recent study, the incidence of adult herpes simplex encephalitis (HSE) in Denmark was estimated to be 4.64 cases per million people per year, corresponding to 27 cases annually [4]. Symptoms of HSE include viral symptoms, headache, decreased consciousness, cognitive changes, and seizures, and HSE is associated with severe disability, neurological sequelae, and relapsing of symptoms [5,6,7].

Approximately 20% of HSE patients experience a flare-up in symptoms after completing antiviral treatment, without detectable levels of HSV in the cerebrospinal fluid (CSF) [8,9]. In recent years, it has become clear that viral encephalitis can trigger an autoimmune response leading to secondary autoimmune encephalitis (AE) [10,11]. In relation to this, an increasing number of HSE patients developing post-viral *N*-Methyl-D-Aspartate receptor autoimmune encephalitis (NMDARE) have been reported [9,12,13,14,15]. The presence of NMDAR autoantibodies in post-HSE patients could be linked to impaired recovery and relapsing symptoms [9,16,17]. As a result, the suspected HSE relapse cases, with no detectable HSV levels in CSF, could potentially be a secondary autoimmune response resulting in NMDARE, rather than a viral flare-up [8]. Finding and recognizing post-HSE NMDARE patients continues to be challenging. Due to overlapping symptoms and variation in the clinical presentation of post-HSE NMDARE, it may be difficult to distinguish between a relapse of HSE and secondary NMDARE based on the clinical picture alone [16,18]. This may lead to delayed diagnosis and treatment.

Most of the HSE patients developing relapsing neurological symptoms or flare-ups are children who often present a characteristic encephalopathy with choreoathetosis, despite a negative CSF viral test [9,11,19,20,21,22]. However, the clinical presentation and disease course appear to be different in teenagers and adults compared to children [9,19]. As the pathogenic effect of NMDAR autoantibodies is caused by receptor suppression and not cellular damage like in HSE, NMDARE is reversible with correct and timely treatment [16,23]. Therefore, early diagnosis and treatment are crucial and associated with a better outcome [24].

In Denmark, the incidence of NMDARE is 1.75 per million people per year [25]. Post-HSE NMDARE appears unrecognized in Denmark; the incidence is not known, and the Danish patient cohort has not been described. In the present study, we therefore aimed to retrospectively identify and describe Danish patients with post-HSE NMDARE using our national database of AE patients from 2009 to 2019 [25]. In addition, we went through two regional infectious encephalitis databases containing information on adult patients in the Region of Southern Denmark diagnosed from 2001 to 2013 and 2015 to 2021. CSF from the identified HSE patients was tested for NMDAR autoantibodies if available. Finally, we performed a literature review of post-HSE NMDARE cases. The Danish cases were compared to the international cases regarding age, symptoms, treatments, and outcomes.

## 2. Materials and Methods

### 2.1. Review of the Literature

To find post-HSE NMDARE cases in the literature for comparison, a literature search of PubMed was undertaken from 2009 to 2021, using the following search terms: “Anti-*N*-Methyl-D-Aspartate Receptor Encephalitis”; “NMDA receptor encephalitis”; “NMDAR Encephalitis”; “Anti-NMDAR Encephalitis”; “Anti NMDA Receptor Encephalitis”; “*N*-methyl-D-aspartate receptor antibodies”; “Encephalitis, Viral”; or “Encephalitis, Herpes Simplex”. A total of 60 publications were screened by two of the authors and abstracts reviewed. The publication exclusion criteria included: non-full text publications, non-English manuscripts, patients not fulfilling diagnostic criteria for encephalitis or without originally confirmed HSV DNA within the CSF by polymerase chain reaction (PCR), reports without individual patient data for analysis, and cases with missing data for analysis. A total of 31 publications were finally included and estimated suitable for further analysis, comprising 80 patients (Figure 1).

All the cases were reviewed for data regarding diagnostics, including imaging studies, electroencephalogram (EEG) and CSF findings, time from viral encephalitis caused by HSV to proven NMDAR autoantibodies, symptoms of NMDARE, treatment, and estimated outcome.

The post-HSE NMDARE cases included were divided based on age of onset into: children (<18 years) and adults (≥18 years) (Figure 1).

### 2.2. Identification of Danish Cases of Post-HSE NMDARE

As part of a previous study, we established a national database of Danish patients diagnosed with AE from 2009 to 2019 [25]. Danish anti-NMDAR IgG-positive cases with preceding HSE were retrospectively identified using this database. The inclusion criteria were confirmed HSV DNA in the CSF identified by PCR, an initial disease course fulfilling criteria for encephalitis, and an either biphasic or prolonged disease course fulfilling diagnostic criteria for NMDARE with detection of NMDAR IgG in CSF.

In addition, we reviewed two registries of adult patients with confirmed infectious encephalitis in the Region of Southern Denmark. The first registry (2001–2013) comprised 33 patients diagnosed with HSE or meningoencephalitis when using the International Classification of Disease-10 diagnosis coding system. The second registry (2015–2021) comprised 89 patients diagnosed with HSE or Varicella Zoster Virus encephalitis. Data were not available from the year 2014. We only included patients diagnosed from 2009 to 2021, with verified polymerase chain reaction (PCR) HSV findings in CSF.

The data were retrospectively assessed according to: (i) symptoms (onset and during disease course); (ii) paraclinical findings (neuroimaging, EEG, and CSF); (iii) treatment; (iv) course of disease; and (v) outcome. Unavailable or insufficient data were categorized as missing data.

HSE patients were divided into a group with a prolonged or relapsing disease course and a group without. The following criteria were used: (1) No significant improvement six weeks after symptom onset; (2) recovery followed by the worsening of symptoms; (3) onset of new neurological symptoms within a year from symptom onset; and (4) exclusion of other causes such as infection.

Not all the medical charts had adequate documentation regarding laboratory findings such as CSF results; some were only mentioned as “normal”. To avoid the exclusion of these normal findings, a mean of the reference interval was used (2.5 WBC/mm^2^ for leukocytes and 0.55 g/L for protein in CSF).

Diagnostic tests performed for neuronal autoantibodies related to HSE admission and one year forward were identified within the medical charts. We divided all the tests performed and the patients tested into two groups: positive and negative for NMDAR autoantibodies.

The outcome was assessed using the modified Rankin Scale (mRS) and a good outcome was considered as mRS scores ≤ 2.

### 2.3. Consensus Criteria for NMDARE

Patients were evaluated using the international consensus criteria (Graus criteria) for probable and definite NMDARE [18]. One criterion regarding the exclusion of other disorders was not included in our application of probable NMDARE during first admission, as all patients had HSE. Patients fulfilling the criteria for probable and definite NMDARE were compared to those not fulfilling the criteria.

### 2.4. Testing of Residual Samples for Autoantibodies

Residual CSF samples from patients diagnosed with HSE were obtained from clinical biobanks at the Departments of Clinical Immunology and Infectious Diseases at Odense University Hospital, if available for research.

If samples were obtained within one year of HSE symptom onset, and in relation to relapsing or prolonged symptoms, they were analyzed for autoantibodies at the Department of Clinical Immunology (national test center for neural autoantibody testing). A standard commercial cell-based assay (CBA, Euroimmun, Lübeck, Germany), testing for autoantibodies against NMDAR, α-amino-3-hydroxy-5-methyl-4-isoxazolepropionic acid receptor 1 + 2 (AMPAR 1 + 2), Contactin-associated protein-like 2 (Caspr2), *γ*-aminobutyric acid B receptor (GABABR), and Leucine-rich Glioma Inactivated 1 (LGI1) was used. The analysis was done blinded to clinical data. Blood samples were not tested, as CSF is more suitable for detecting anti-NMDAR IgG [26,27].

### 2.5. Incidence of Post-HSE NMDARE

The incidence of post-HSE NMDARE was estimated based on patients with definite post-HSE NMDARE in the Region of Southern Denmark. Since post-viral autoimmunity was first reported and widely recognized from 2013, our calculations are based on numbers from 2013 and onwards. The number of cases was divided by the mean population in the Region of Southern Denmark from 2013 to 2021 (1,212,122 million) (from Statistics Denmark [28]) and the time period of 7.67 years (1 January 2013 to 1 October 2021; 2014 not included).

### 2.6. Ethical Considerations

The study was approved by the Regional Committees on Health Research Ethics for Southern Denmark (3-3013-3124/1), the data protection agency and the Danish Patient Safety Authority (3-3013-2579/1). All autoantibody testing was performed on stored excess CSF from the patients’ diagnostic work-up. The patients did thus not undergo new diagnostic procedures.

### 2.7. Statistics

A statistical analysis was performed using GraphPad Prism 6 software (GraphPad) and IBM SPSS 28 statistics software (Version: 28.0.0.0 for Windows).

For numerical variables, a median and range were calculated. For categorical variables, data were given as % (*n*/N). The demographics, clinical features, medical work-up, treatment, and outcome were analyzed using Fisher’s exact test, unpaired *t* test, and Mann-Whitney U test, when appropriate. A *p*-value < 0.05 was considered statistically significant.

## 3. Results

### 3.1. Danish Patients with Post-HSE NMDARE

From 2009 to 2019, eight patients were found to develop NMDARE after HSE infection in Denmark. All the patients have previously been reported in a paper describing the National Danish NMDARE cohort [25]. The median age was 61 years (range 46–74 years) with no female/male predominance. Interestingly, we found no children among our post-viral NMDARE patients.

### 3.2. Cases from Literature

Between 2009 and 2021, a total of 80 patients were identified with post-HSE NMDARE in the literature; 53 children (<18 years) and 27 adults (≥18 years) (Figure 1).

In the children’s cohort, the median age at diagnosis was 1.3 years (range 0–17 years) compared to a median age of 51 years in adults (range 24–84 years). No differences in sex were found. The median time between the onset of HSE and diagnosis of NMDARE was shorter in children; 27 days (range 4–65 days) than it was in adults; 61 days (range 7–365 days) (Table 1). The presentation of symptoms during NMDARE was different in children compared to adults. Movement disorders were the most frequent symptom in children, occurring in 89% (*n* = 47) compared to only 22% (*n* = 6) for adults. Furthermore, altered consciousness; 62% (*n* = 33), sleep disturbances; 40% (*n* = 21), and autonomic dysfunctions; 36% (*n* = 19), occurred more often in children compared to adults (Table 1).

In contrast, psychiatric symptoms, behavioral change, and cognitive impairment were the most frequent symptoms in the adult cohort, all occurring in 59% (*n* = 16) of the patients, with the latter two significantly more often in adults than children (Table 1).

The differences in the clinical phenotype between children and adults with post-HSE NMDARE in the literature are illustrated in Figure 2.

At the time of NMDARE diagnosis, the CSF findings were abnormal in most cases with a median leukocyte count in children of 29 WBC/mm^2^ (range 0–185) compared to 13 WBC/mm^2^ in adults (range 1–480). Protein levels were elevated in 41% (*n* = 7) of children and 72% (*n* = 13) of adults and when tested for oligoclonal bands they were mostly present in both groups (100% children, 77% adults) (Table 1).

EEG changes during NMDARE disease were seen in all patients who underwent testing, with paroxysmal activity occurring less frequently in children; 14% (*n* = 1) compared to adults; 60% (*n* = 6). Slowing activity was found equally in children and adults (57% (*n* = 4) vs. 50% (*n* = 5)) (Table 1). Additional MRI changes to previously found abnormalities during the HSE disease course occurred in 26% of children (*n* = 6) and 37.5% (*n* = 6) of adults (Table 1).

Children were more often treated with immunosuppression compared to adults (96% (*n* = 51); 59% (*n* = 16)). We found no difference in the use of first- and second-line treatment in literature cases. Maintenance therapy was given more often in children; 24% (*n* = 12) compared to adults; 12.5% (*n* = 2) (Table 1).

The outcome measured by mRS showed a poorer prognosis in children with a median mRS of 3 (range 0–6) compared to a median mRS score of 2 (range 0–6) in adult cases. Thus, 67% (*n* = 35) of children compared to 41% (*n* = 9) of adults were categorized as having a poor outcome.

An overview of the 80 cases and their references can be found in Appendix A.

### 3.3. Are Possible Adult Post-HSE NMDARE Cases Overlooked in Denmark?

#### 3.3.1. Identifying HSE Patients in the Region of Southern Denmark

Based on previous international reports [9,11,19,20,21], we wondered if adult HSE NMDARE cases were still overlooked in Denmark. When running through the two infectious encephalitis databases of the Region of Southern Denmark, we identified 44 HSE patients between 2009 and 2021 (no data were available for the year 2014) (Figure 3).

The median age at diagnosis of HSE was 69 years, with a female:male ratio of 1:1. The majority of patients presented with fever (73%), with a median temperature of 39 °C. Cognitive impairment, speech abnormalities, personality changes, and seizures were the most abundant symptoms (95%, 66%, 53%, and 52%, respectively). The median WBC/mm^2^ in CSF was 69 leukocytes (range 2–790). Eight of the 44 patients (18%) fulfilled our criteria for having a prolonged (*n* = 1) or a relapsing HSE disease (*n* = 7).

Overall, ten patients (23%) were tested for AE autoantibodies during their HSE disease course (including both first and re-admissions). In total, two patients tested positive for NMDAR antibodies, both during a relapse. These two patients were already identified and included in the National NMDARE cohort. No patients tested positive for NMDAR autoantibodies at first admission.

A detailed overview of the presenting symptoms, diagnostic work-up, and outcomes of the HSE patients can be found in Appendix A.

A higher percentage of the patients had their CSF tested for oligoclonal bands and/or IgG index (75% (*n* = 3) vs. 33% (*n* = 1)) and NMDAR autoantibodies (50% (*n* = 4) vs. 25% (*n* = 2)) at second admission compared to first admission (Appendix A).

Additionally, material on the first and second admission of the eight patients with prolonged/relapsing HSE disease course is provided in Appendix A.

#### 3.3.2. Screening HSE Patients for Probable Post-Viral NMDARE Using Clinical Diagnostic Criteria

A total of 16 (36%) of the HSE patients fulfilled the criteria for probable or definite NMDARE at either first and/or second admission (Table 2).

Though not significant, we observed a tendency towards a higher proportion of patients fulfilling the Graus criteria presenting a prolonged/relapsing disease course, compared to patients not fulfilling the criteria (31% (*n* = 5) vs. 11% (*n* = 3)) (Table 2). Furthermore, patients fulfilling the Graus criteria had more severe HSE disease courses, with a lower Glasgow Coma Scale (GCS) during first admission compared to patients not fulfilling them (median GCS of 6 (range 2–13) vs. 13 (range 3–15), *p* < 0.0001)) (Table 2).

There was a tendency towards a higher prevalence of testing CSF for AE autoantibodies in the probable/definite NMDARE group (37% (*n* = 6) vs. 14% (*n* = 4)) (Table 2). The length of first admission was longer in the probable/definite NMDARE group, with a median of 30 days vs. 17 days in the group not fulfilling the criteria (*p* = 0.003) (Table 2).

#### 3.3.3. Retrospective Screening of HSE Patients for Autoantibodies

An overview of patients with HSE, how many had a relapsing/prolonged disease course, and the frequency of AE autoantibody testing performed each year from 2009 to 2021 can be seen in Figure 4.

A total of 16 AE autoantibody tests were performed, distributed on 14 patients (Table 3). Of the 16 tests, 10 were analyzed during HSE admission and 6 were tested retrospectively as part of this study, as they had excess CSF stored in the biobank from their HSE admission. Two of the 16 patients (13%) were found to harbor NMDAR antibodies during relapse admission, and additionally one of the six (17%) samples tested positive (Table 3). Thus, in total three of the 14 patients (21%) tested for AE autoantibodies tested positive for NMDAR antibodies after HSE. None harbored other AE autoantibodies than NMDAR. All of the three NMDAR antibody positive patients displayed a relapsing HSE disease course.

When comparing the NMDAR IgG-positive patients (*n* = 3) with the NMDAR IgG negative patients (*n* = 11), there was a clear difference regarding time from symptom onset to lumbar puncture, with a median of 90 days (range 35–147 days) in those who tested positive vs. 7.5 days (range 1–35) in those who tested negative (Table 3).

#### 3.3.4. Incidence of Post-HSE NMDARE in Denmark

The incidence of post-HSE NMDARE in the Region of Southern Denmark, based on identified definite cases, was 0.17/million people/year.

Among these patients, 39% fulfilled the diagnostic criteria for probable or definite NMDARE. When based solely on definite identified cases, 7% of HSE patients in the Region of Southern Denmark were diagnosed with post-HSE NMDARE.

### 3.4. Comparison of Danish Adult Post-HSE NMDARE Cases to Adult Literature Cases

Combining the eight cases of post-HSE NMDARE already identified with the one additional patient found by testing spare CSF, we identified nine post-HSE NMDARE patients in Denmark from 2009 to 2021. All were adults, and none were children.

When comparing these nine Danish adult cases to adult cases described in the literature, we found a similar male:female ratio, but Danish patients where significantly older (62 vs. 51 years, *p* = 0.0316) (Table 4). We found no difference in time from HSE to diagnosis of NMDARE.

Memory impairment and cognitive impairment were the most frequent symptoms in the Danish cohort, both occurring in 78% (*n* = 7) of the cases, followed by psychiatric symptoms in 67% (*n* = 6). In comparison, these symptoms were present in 30% (*n* = 8), 59% (*n* = 16) and 59% (*n* = 16) of the literature cases, respectively (Table 4). The CSF findings during both the HSE and the NMDARE disease course were similar between Danish and literature-reported patients (Table 4). Correspondingly, the EEG and Brain MRI findings were similar between the two groups. All patients in the Danish cohort underwent MRI compared to only 59% (*n* = 16) of patients in the literature cohort, with 44% (*n* = 3) and 37% (*n* = 6) having MRI changes in addition to previous HSE changes (Table 4).

The use of first- and second-line treatment was lower for the Danish cases compared to the literature cases, as 67% (*n* = 6) of Danish cases and 100% (*n* = 16) of literature cases received first-line treatment, and none of the Danish cases and 56% (*n* = 9) of the literature cases received second-line treatment. Maintenance therapy was given more frequently in the Danish cases compared to the literature cases (67% vs. 12.5%) (Table 4).

The outcome was slightly worse in the Danish cases, with a median mRS of 3 (range 1–5) compared to a median mRS score of 2 (range 0–6) in the literature cases. However, one patient (4.5%) in the literature cohort died. Fifty-six percent of the Danish cases (*n* = 5) and 41% of literature cases (*n* = 9) were categorized as having a poor outcome (Table 4).

## 4. Discussion

In this study, our aims were to identify and describe all Danish post-HSE NMDARE patients, investigate whether Danish patients are overlooked, and compare Danish patients to patients described in the literature. From 2009 to 2019, we found eight Danish post-HSE NMDARE patients. All eight patients were identified through our national AE cohort, which is based on every positive autoantibody test result in the mentioned time period, including both children and adults [25]. Nevertheless, all the patients identified were adults, as no children with post-HSE NMDARE were found. In the literature review, 66% of post-HSE NMDARE cases were children, and both HSE and NMDARE are well recognized diseases in children [9,10,20,21,24]. There might be several explanations for the lack of children with post-HSE NMDARE in Denmark. Firstly, our national cohort also comprises fewer children (20%) than other international described cohorts, suggesting that children are generally less frequently tested for NMDARE autoantibodies in Denmark [25]. Secondly, it can be difficult to determine whether relapse post-HSE is due to a viral or autoimmune cause, as symptoms may vary between patients, but also between children and adults [9,10,11]. Thus, diagnostic clues on when to suspect post-viral autoimmunity, especially in children, are an important tool for pediatricians. The findings in our literature review suggest that children with post-HSE NMDARE most often present with a clinical phenotype of abnormal movements, altered consciousness, and sleep disturbances. These findings are in line with the findings from a recent, more comprehensive literature review [29].

Based on the severe lack of children diagnosed with post-HSE NDMARE in Denmark, it would have been preferable to examine a pediatric HSE cohort, to evaluate how many potential post-HSE NMDARE cases were overlooked. Unfortunately, we did not have access to a pediatric HSE cohort.

We then examined whether adult post-HSE NMDARE patients were still overlooked in Denmark. We identified all the HSE patients in one region of Denmark (the Region of Southern Denmark). We registered how many had a disease course that could raise suspicion of post-viral autoimmunity, how many were tested for autoimmune encephalitis, and how many fulfilled the Graus criteria for probable/definite NMDARE, and in cases with leftover CSF in the national biobank, we retested for autoantibody presence.

Out of 44 HSE patients in the Region of Southern Denmark, eight (18%) had a disease course suggestive of post-viral autoimmunity (prolonged or relapsing disease course), ten (23%) were tested for autoimmune encephalitis during hospitalization, and 16 (36%) fulfilled the Graus criteria for probable or definite NMDARE. Unfortunately, only six (14%) of the 44 patients had leftover CSF to be retrospectively tested. However, even with this limited number of patients to be reevaluated, still one out of the six (17%) was found positive for NMDAR autoantibodies in the CSF, and thus initially overlooked.

It was previously described that 20% of HSE patients experience relapsing or ongoing symptoms after antiviral treatment, despite a negative HSV PCR in CSF [8]. This correlates with our findings of 18% (8/44) of patients with a relapsing/prolonged HSE disease course. A study implied the importance of testing patients with prolonged, worsening, or relapsing symptoms after HSE for autoantibodies [10]. In line with this, out of our eight patients, six had CSF tested for autoantibodies (two at first admission, four at relapse admission), of which two were positive. Having this association in mind, comparing HSE patients with a relapsing/prolonged disease course to those without may predict possible prognostic factors for developing NMDARE secondary to HSE. However, the only significant difference in the clinical phenotype at first admission was decreased consciousness (*p* = 0.019). Nevertheless, this might be a risk factor for developing a relapsing/prolonged disease course and thus post-viral autoimmunity. One other study has reported a higher percentage of decreased consciousness as well as motor deficits and aphasia amongst the relapse group at first admission compared to patients with no relapse, though not significant [9].

Furthermore, we wanted to investigate whether fulfilling the criteria for probable NMDARE at admission was a risk factor for developing post-HSE NMDARE. Overall, 36% of HSE patients fulfilled the criteria for probable or definite NMDARE. However, only two of the eight patients (25%) with a relapsing/prolonged disease course fulfilled the probable NMDARE criteria. This highlights the problem in using these criteria in post-HSE NMDARE patients and indicates that these patients do not always present with classic NMDARE symptoms or paraclinical findings. Another problem with applying the Graus criteria at first admission is that it was not possible to rule out “other disorders” because all the patients had HSE. Furthermore, HSE is characterized by many of the same symptoms and clinical features, which can lead to an overestimation of cases with probable NMDARE.

When comparing the Danish post-HSE patients with the literature, we generally found a similar clinical phenotype as previously reported. Adult post-HSE NMDARE patients present a less severe disease course than idiopathic NMDARE patients, dominated by symptoms of cognitive impairment, behavioral change, and psychiatric symptoms. Patients develop post-viral autoimmunity at a median of approximately 1–2 months after the HSE disease course, and generally present light inflammatory CSF (<30 WBC/mm^2^) when compared to their HSE disease course (>100 WBC/mm^2^). Interestingly, additional findings on MRI can be seen in up to 40% of patients. These findings are all in line with previously described cohorts [29].

However, the Danish patients differed from the literature cases when evaluating treatment strategy. Danish patients more often received no treatment (only 67% received first-line therapy). Additionally, when treated, none received treatment escalation with second-line therapy. Despite this, no significant difference in outcome was seen. However, the outcome must be interpreted with caution, as it was only measured using the mRS and we had no information on the follow-up time for the literature cohort due to the retrospective nature of the study.

Interestingly, our and one recent literature review revealed that children seem to have a worse outcome than adults [29]. It can be speculated that the underdevelopment of the nervous system in children and adolescents affects normal CNS functioning more than in adult-onset episodes [30].

As we reevaluated HSE patients in only one region of Denmark, the most precise incidence we could estimate was the incidence of post-HSE NMDARE in that region. Thus, we found an incidence of post-HSE NMDARE in the Region of Southern Denmark of 0.31/per million people/per year (from 2013 to 2021).

Of the identified 44 HSE patients in the region, three (7%) were diagnosed with post-HSE NMDARE. This is a clear discrepancy to previously reported studies, which report more than double the percentage of patients (17–24%) [9,17]. One explanation for this discrepancy is of course the lack of children and adolescents in our Danish cohort compared to other cohorts and literature reviews [9,31]. An additional explanation for our relatively low incidence of post-HSE NMDARE could be due to adult patients still being overlooked. This is highlighted by the fact that we, by selection of clinical phenotype alongside disease course and re-testing of only six leftover CSF samples, were able to identify one additional patient. It is likely that other patients, even in this regional cohort of 44 cases and in the rest of Denmark, are still overlooked, due to a lack of autoantibody testing. This indicates low awareness among clinicians.

### Limitations

The main limitation of this study is that HSE and consequently post-HSE NMDARE is a relatively rare disease, resulting in a small study population size, compromising statistical power. Furthermore, both the heterogeneity of the studies included in the literature search and the retrospective nature of our study limit the level of detail in clinical examination and the availability of information, resulting in missing data and increasing the risk of misclassification bias. 

We have not identified any children with post-HSE NMDARE in Denmark so far. A limitation of our study was that we were only able to re-examine adult HSE patients in one region of Denmark, with no available pediatric HSE cohort. Furthermore, the availability of leftover CSF samples to retrospectively test for autoantibodies in the regional HSE cohort was limited.

## 5. Conclusions

Overall, the Danish adult post-HSE NMDARE cohort resembles those reported in the literature, with the exception of an older adult cohort and less aggressive treatment strategy in Denmark.

A lower percentage of HSE patients were found to develop post-HSE NMDARE compared to previous studies, suggesting that adult patients are still overlooked. No children/adolescents with post-HSE NMDARE have been identified in Denmark until 2021. As the majority of the post-HSE NMDARE patients in the literature have been reported to be children, these must be immensely overlooked or misdiagnosed. Our findings underscore the need to address the diagnostic and therapeutic challenges involved in managing pediatric post-HSE cases in particular. The absence of pediatric cases in Denmark highlights a potential oversight in the healthcare system, necessitating immediate action to ensure that young patients receive an accurate and timely diagnosis and treatment. Physicians should be better equipped to recognize the youngest patients. Clinical clues such as involuntary movements, decreased consciousness, and sleep disturbances in post-HSE in children should raise suspicion about post-viral NMDARE.

## Figures and Tables

**Figure 1 biomedicines-12-01953-f001:**
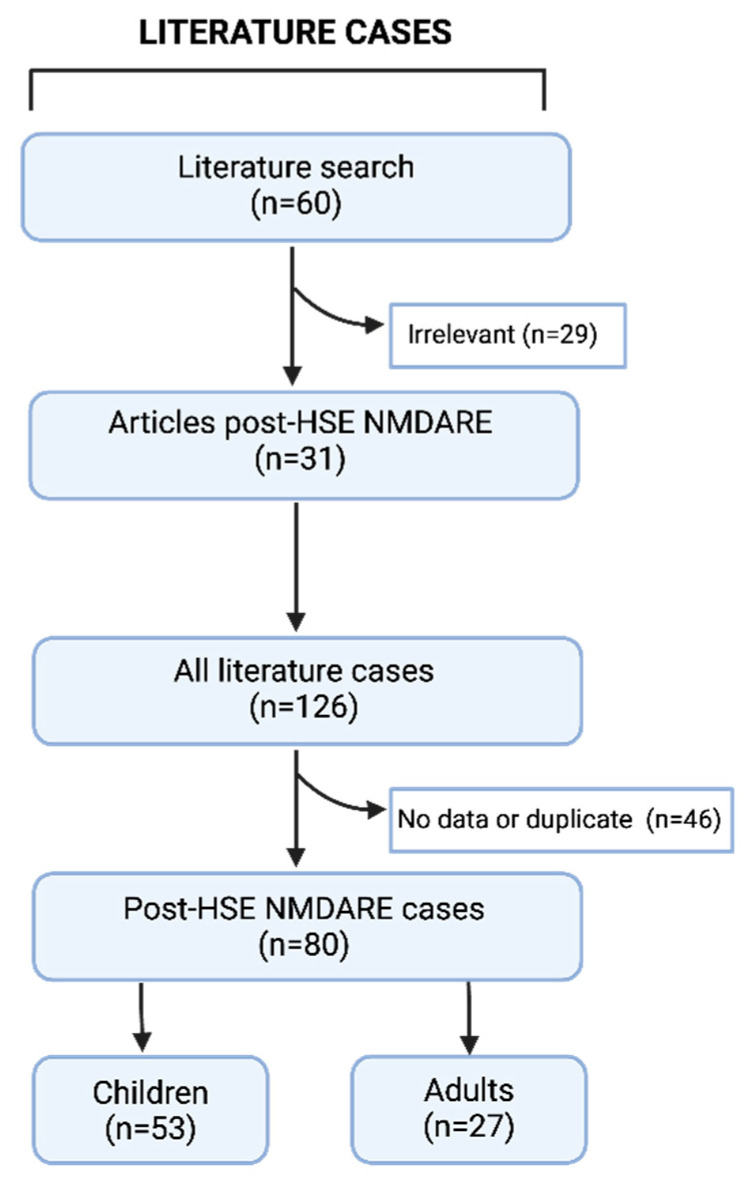
Sixty papers were identified; 29 were considered irrelevant, resulting in 31 papers providing 126 relevant cases. Due to insufficient data/duplicate cases, 46 cases were excluded. A total of 80 cases were included and divided based on age of onset into: children (<18 years) and adults (≥18 years). Abbreviations: post-HSE NMDARE, post-Herpes Simplex Virus 1 *N*-methyl-D-Aspartate Receptor Encephalitis.

**Figure 2 biomedicines-12-01953-f002:**
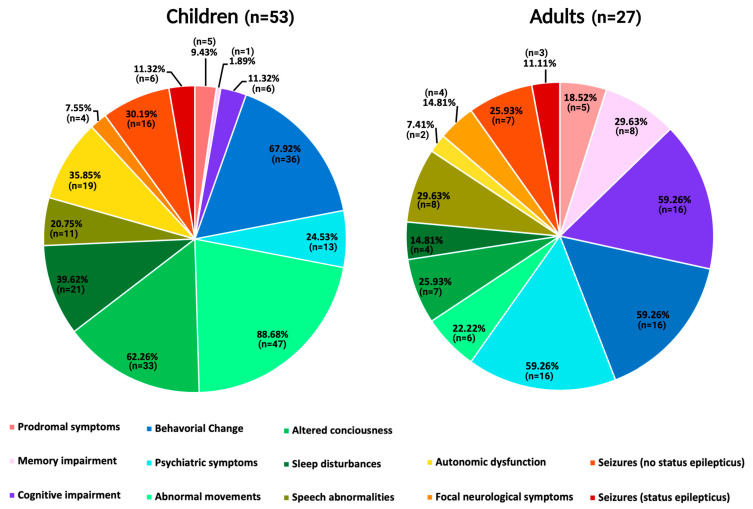
Pie chart displaying the distribution of post-HSE NMDARE symptoms in children and adult literature cases. Abbreviations: HSE, herpes simplex encephalitis. NMDARE, *N*-Methyl-D-Aspartate receptor encephalitis.

**Figure 3 biomedicines-12-01953-f003:**
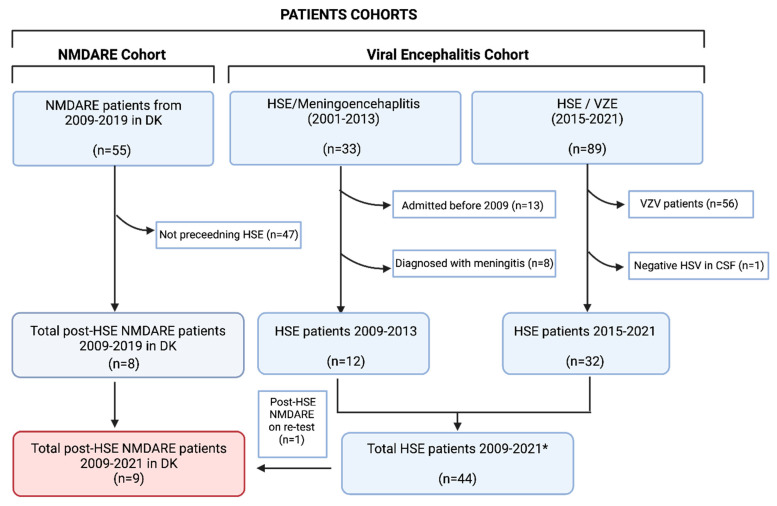
Overview of screening of national and regional databases for post-HSE NMDARE patients. The Danish National NMDARE Cohort (left) was screened for patients with preceding HSE; a total of eight patients developed NMDARE after HSE infection. Two regional infectious encephalitis databases (right) were screened for patients with HSE; 44 patients were found. Of these, one additional patient developing NMDARE after HSE was identified. Abbreviations: HSE, Herpes Simplex Virus Encephalitis. NMDARE, *N*-Methyl-D-Aspartate receptor Encephalitis; VZE, Varicella Zoster Virus Encephalitis; VZV. Varicella Zoster Virus. * No data available for 2014.

**Figure 4 biomedicines-12-01953-f004:**
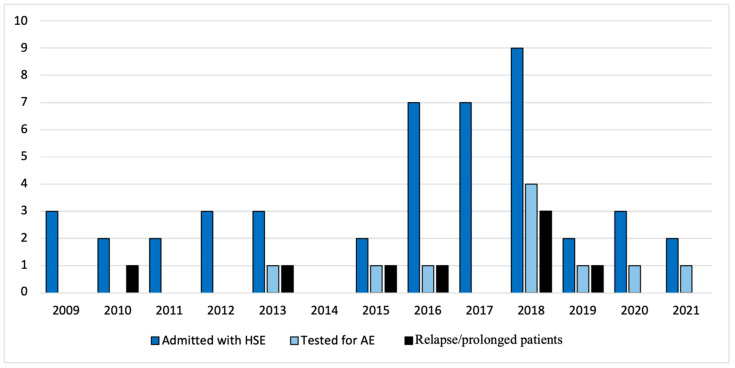
Bar chart illustrating patients admitted with HSE (blue), patients with relapsing/prolonged disease course (black) and the total number of tests for AE autoantibodies in CSF performed (light blue) in the Region of Southern Denmark 2009–2021. Abbreviations: AE, Autoimmune Encephalitis; *HSE*, Herpes Simplex Encephalitis.

**Table 1 biomedicines-12-01953-t001:** Literature Search.

	Literature Post-HSE NMDARE Cases			*p*-Value
	All Cases *n* = 80 (%)	Children *n* = 53 (%)	Adults *n* = 27 (%)	
**Demographics**				
Sex, Female	42 (52)	28 (53)	14 (52)	>0.9999
Median Age, years (range) *	3 (0.1667–84)	1.33 (0.1667–17)	51 (24–84)	<0.0001 ****
**Time HSV1 to NMDAR Ab *n*, (%)**	***n* = 28 (35)**	***n* = 16 (30)**	***n* = 12 (44)**	
Median days (range)	29.00 (4–365)	26.5 (4–65)	60.62 (7–365)	0.1993
**Clinical presentation**				
Prodromal symptoms	10 (12)	5 (9)	5 (18)	0.2925
Memory impairment	9 (11)	1 (2)	8 (30)	**0.0005 *****
Cognitive impairment	22 (27)	6 (11)	16 (59)	**<0.0001 ******
Behavioral change	52 (65)	36 (68)	16 (59)	0.08665
Psychiatric symptoms	29 (36)	13 (24)	16 (59)	0.3473
Abnormal movements	53 (66)	47 (89)	6 (22)	**<0.0001 ******
Altered consciousness	40 (50)	33 (62)	7 (26)	**0.0041 ****
Sleep disturbances	25 (31)	21 (40)	4 (15)	**0.0398 ***
Speech abnormalities	19 (24)	11 (21)	8 (30)	0.4133
Autonomic dysfunction	21 (26)	19 (36)	2 (7)	**0.0068 ****
Focal neurological symptoms	8 (10)	4 (8)	4 (15)	0.4324
Seizures (no status epilepticus)	23 (29)	16 (30)	7 (26)	>0.9999
Seizures status epilepticus	9 (11)	6 (11)	3 (11)	>0.9999
**CSF findings in HSE course (%)**	***n* = 34 (42)**	***n* = 19 (36)**	***n* = 15 (56)**	
Abnormal CSF, *n* (%)	33 (97)	18 (95)	15 (100)	>0.9999
Pleocytosis, *n* (%)	32 (94)	18 (95)	14 (93)	>0.9999
Median WBC/mm^3^ (range) ***	104 (0–920)	64 (2–920)	175 (0–730)	0.1093
Protein elevation, *n* (%)	20/29 (69)	7/15 (47)	13/14 (93)	**0.0142 ***
Median protein g/L (range) ****	0.91 (0.34–4.44)	0.77 (0.34–4.44)	0.94 (0.52–3)	0.3673
OCB presence, *n* (%)	NA	NA	NA	-
**CSF findings in NMDARE course (%)**	***n* = 35 (44)**	***n* = 17 (32)**	***n* = 18 (678)**	
Abnormal CSF, *n* (%)	34 (97)	16 (94)	18 (100)	0.4857
Pleocytosis, *n* (%)	28 (80)	13 (76)	15 (83)	0.6906
Median WBC/mm^3^ (range) **	20 (0–480)	29 (0–185)	13 (1- 480)	0.5898
Protein elevation, *n* (%)	20 (57)	7 (41)	13 (72)	0.0922
Median protein g/L (range)	0.82 (0.39–2.56)	0.78 (0.45–2.49)	0.88 (0.39–2.56)	0.7133
OCB presence, *n* (%)	11/13 (85)	4/4 (100)	7/9 (78)	0.4705
**EEG in NMDARE course**	***n* = 17 (21)**	***n* = 7 (13)**	***n* = 10 (37)**	
Abnormal EEG, *n* (%)	14 (82)	5 (71)	9 (90)	0.5368
EEG paroxysmal, *n* (%) *****	7 (41)	1 (14)	6 (60)	0.134
EEG slowing, *n* (%) *****	9 (53)	4 (57)	5 (50)	>0.9999
EEG delta brush, *n* (%)	0 (0)	0 (0)	0 (0)	-
**MRI changes additional in NMDARE**	***n* = 39 (49)**	***n* = 23 (43)**	***n* = 16 (59)**	
Abnormal, *n* (%)	12 (30.77)	6 (26.09)	6 (37.5)	0.498
**Treatment (%)**	***n* = 73 (91)**	***n* = 53 (100)**	***n* = 20 (74)**	
Received any treatment, *n* (%)	67 (92)	51 (96)	16 (80)	**0.0443 ***
Treatment—1st line ^a^, *n* (%)	67 (100)	51 (100)	16 (100)	>0.9999
Treatment—2nd line ^b^, *n* (%)	39 (58)	30 (59)	9 (56)	>0.9999
Treatment—Maintenance, *n* (%)	14 (21)	12 (23)	2 (12)	0.4898
**Outcome mRS, *n* (%)**	***n* = 74 (92)**	***n* = 52 (98)**	***n* = 22 (81)**	
Mean mRS (range)	2.77 (0–6)	3.04 (0–6)	2.14 (0–6)	**0.0352 ***
Good outcome at last follow-up (mRS 0–2), *n* (%)	30 (40)	17 (33)	13 (59)	**0.0418 ***
Poor outcome at last follow-up (mRS 3–6), *n* (%)	44 (59)	35 (67)	9 (41)	**0.0418 ***
**Dead, *n* (%)**	2 (3)	1 (2)	1 (5)	0.5091

* In one patient, age was not described. ** Information on three children and one adult was not available. *** Information on one child and one adult was not available. **** Information on six children and five adults was not available. ***** One patient had both slowing and paroxystic activity. ^a^ 1st line treatment: High-dose intravenous steroids in combination with intravenous immunoglobulin and/or therapeutic plasma exchange. ^b^ 2nd line treatment: Rituximab or Cyclophosphamide. Abbreviations: Ab, Antibodies; CSF, Cerebrospinal Fluid; EEG, Electroencephalogram; HSV1, Herpes Simplex Virus 1; HSE, Herpes Simplex Virus 1 Encephalitis; MRI, Magnetic Resonance Imaging; mRS, modified Rankin Scale; NMDAR, *N*-Methyl-D-Aspartate Receptor; NMDARE, *N*-Methyl-D-Aspartate Receptor Encephalitis; OCB, Oligoclonal Bands; WBC, White Blood Cell.

**Table 2 biomedicines-12-01953-t002:** Characteristics of HSE patients respectively fulfilling or not fulfilling the Graus criteria.

	Probable + Definite (*n* = 16)	Not Fulfilling Criteria (*n* = 28)	*p*-Value
**Characteristics**			
Sex. (male), % (*n*/N)	50 (6/16)	54 (15/28)	>0.9999
Age, years, median (range)	73 (35–83)	65 (25–88)	0.0733
Prolonged/relapse, % (*n*/N)	31 (5/16)	11 (3/28)	0.1172
GCS at admission, median (range)	6 (2–13)	13 (3–15)	**<0.0001 ******
**Autoantibody Testing**			
Tested for NMDAR-abs, % (*n*/N)	37 (6/16)	14 (4/28)	0.1331
Positive NNMDAR-abs, % (*n*/N)	33 (2/6)	0 (0/4)	0.4667
**Timeline, days, median (range)**			
Admission length	30 (12–82)	17 (6–52)	**0.0030 ****
Symptoms to admission	3 (0–14)	2 (0–21) (n = 25)	0.7526
Symptoms to HSE treatment	5 (3–29)	4 (0–22) (*n* = 26)	0.1966

Abbreviations: Ab, Antibodies; GCS, Glasgow Coma Scale; NMDAR, *N*-Methyl-D-Aspartate Receptor. ** and **** refer to the level of significance.

**Table 3 biomedicines-12-01953-t003:** Characteristics of samples for autoantibody testing (CSF).

Time of Testing, % (*n*/N)	Autoantibody Tests *n* = 16	Test Positive *n* = 3	Test Negative *n* = 13
At first admission	37 (6/16)	0 (0/3)	46 (6/13)
At relapse/prolonged admission	25 (4/16)	67 (2/3)	15 (2/13)
At re-test (CSF, Biobank)	37 (6/16)	33 (1/3)	38 (5/13)
**Patient Characteristics**	**Patients tested *n* = 14 ***	**Positive *n* = 3**	**Negative *n* = 11**
Time admission to LP, days median (range)	16 (−2–143)	76 (35–143)	7 (−2–76)
Time symptoms to LP, days median (range)	25 (1–147)	90 (35–147)	7 (1–35)
Relapse/prolonged, % (*n*/N)	36 (5/14)	100 (3/3)	18 (2/11)
Probable/definite NMDARE **, % (*n*/N)	43 (6/14)	67 (2/3)	36 (4/11)

* two of the six patients retrospectively tested had already undergone AE autoantibody tests during admission, and where thus tested twice. ** fulfilling criteria at time of autoantibody test. Abbreviations: CSF. Cerebrospinal Fluid; LP. Lumbar Puncture; NMDAR. *N*-Methyl-D-Aspartate Receptor; NMDARE. *N*-Methyl-D-Aspartate Receptor Encephalitis.

**Table 4 biomedicines-12-01953-t004:** Literature cases vs. Danish cases.

	Literature Adults HSE (*n* = 27) (%)	Danish Adults HSE (*n* = 9) (%)	*p*-Value
**Demographics**			
Sex, Female, *n* (%)	14 (52)	4 (44)	>0.9999
Age, median (range) *	51 (24–84)	62 (46–75)	**0.0316 ***
**Time HSE to NMDAR Ab, *n* (%)**	***n* = 12 (44)**	***n* = 9 (100)**	
Days, median (range)	61 (7–365)	45 (25–212)	0.7394
**Symptoms (%)**	***n* = 27 (100)**	***n* = 9 (100)**	
Prodromal symptoms	5 (18)	2 (22)	>0.9999
Memory impairment	8 (30)	7 (78)	**0.0190 ***
Cognitive impairment	16 (59)	7 (78)	0.4378
Behavioral change	16 (59)	4 (44)	0.4697
Psychiatric symptoms	16 (59)	6 (67)	>0.9999
Abnormal movements	6 (22)	4 (44)	0.2262
Altered consciousness	7 (26)	3 (33)	0.6856
Sleep disturbances	4 (15)	1 (11)	>0.9999
Speech abnormalities	8 (30)	5 (56)	0.2347
Autonomic dysfunction	2 (7)	1 (11)	>0.9999
Focal neurological symptoms	4 (15)	5 (56)	**0.0262 ***
Seizures (no status epilepticus)	7 (26)	5 (56)	0.4088
Seizures (status epilepticus)	3 (11)	(0)	>0.9999
**CSF findings in HSE course (%)**	***n* = 15 (56)**	***n* = 8 (89)**	
Abnormal, *n* (%)	15 (100)	8 (100)	-
Pleocytosis, *n* (%)	14 (93)	8 (100)	>0.9999
Median WBC/mm^3^ (range) **	175 (0–730)	102.5 (19–360)	0.2119
Protein elevation, *n* (%)	13/14 (93)	3/6 (50)	0.0609
Median protein g/L (range) ***	0.94 (0.52–3)	0.89 (0.43–1.2)	0.4315
OCB present, *n* (%)	NA	NA	-
**CSF findings in NMDARE course (%)**	***n* = 18 (66.67)**	***n* = 9 (100)**	
Abnormal, *n* (%)	18 (100)	9 (100)	-
Pleocytosis, *n* (%)	15 (83)	9 (100)	0.5292
Median WBC/mm^3^ (range) **	13 (1–480)	23 (7–60)	0.3176
Protein elevation, *n* (%)	13 (72)	8 (89)	0.6279
Median protein g/L (range) ****	0.88 (0.39–2.56)	0.89 (0.64–1.04)	0.9026
OCB present, *n* (%)	7/9 (78)	3/3 (100)	>0.9999
**EEG in NMDARE (%)**	***n* = 10 (37)**	***n* = 7 (78)**	
Abnormal EEG, *n* (%)	9 (90)	6 (86)	>0.9999
EEG paroxysmal, *n* (%) *****	6 (60)	1 (14)	0.1340
EEG slowing, *n* (%) *****	5 (50)	6 (86)	0.3043
EEG delta brush, *n* (%)	0 (0)	0 (0)	-
**MRI changes in NMDARE (%)**	***n* = 16 (59.26)**	***n* = 9 (100)**	
Abnormal, *n* (%)	6 (37)	4 (44)	>0.9999
**Treatment (%)**	***n* = 20 (74)**	***n* = 9 (100)**	
Any treatment, *n* (%)	16 (80)	6 (67)	0.6424
1st line ^a^, *n* (%)	16 (100)	6 (67)	**0.0365 ***
2nd line ^b^, *n* (%)	9 (56)	0 (0)	**0.0078 ****
Maintenance, *n* (%)	2 (12)	6 (67)	**0.0099 ****
**Outcome mRS (%)**	***n* = 22 (81)**	***n*= 9 (100)**	
Median mRS (range)	2 (0–6)	3 (1–5)	0.3159
Good outcome at last follow-up (mRS 0–2), *n* (%)	13 (59)	4 (44)	0.6927
Poor outcome at last follow-up (mRS 3–6), *n* (%)	9 (41)	5 (56)	0.6927
**Dead, *n* (%)**	1 (5)	0 (0)	>0.9999

* age not available in one patient from the literature. ** WBC not available in one patient from the literature. *** Protein level not available in two patients from the literature. **** Protein level not available in one patient from the literature. One Danish patient with no available protein level but stated “normal”. ***** One patient with both slowing and paroxystic activity in the literature. ^a^ 1st-line treatment: High-dose intravenous steroids in combination with intravenous immunoglobulin and/or therapeutic plasma exchange. ^b^ 2nd-line treatment: Rituximab or Cyclophosphamide. Abbreviations: Ab, Antibodies; CSF, Cerebrospinal Fluid; EEG, Electroencephalogram; HSE, Herpes Simplex Virus 1 Encephalitis; MRI, Magnetic Resonance Imaging; mRS, modified Rankin Scale; NMDAR, *N*-Methyl-D-Aspartate Receptor; NMDARE, *N*-Methyl-D-Aspartate Receptor Encephalitis; OCB, Oligoclonal Bands; WBC, White Blood Cell.

## Data Availability

The data and datasets can be made available upon reasonable request.

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
