# Peer review of "Post-Herpetic Anti-NMDAR Encephalitis in Denmark: Current Status and Future Challenges"

_biomedicines, 2024, doi:10.3390/biomedicines12091953_

Round 1

Reviewer 1 Report

Comments and Suggestions for Authors
Dear Authors, 
Thank you very much for submitting your research to our journal. We identified some important issues regarding the methodology and results of your research while reading your paper. These can be found below. 
Major issues: 
-            The search strategy is not clearly defined. In most review articles more than one database should be searched. The inclusion and exclusion criteria for the studies are not clearly defined. Moreover, the authors fail to describe how studies were selected (how many reviewers screened the articles, how did they manage the cases in which reviewers not in consent etc). Did the authors apply any strategy for missing data (e.g. contact the authors) as just exclusion of the studies is not an academic approach?
-            It is not clear how patients in the Danish cohort were included (e.g. inclusion and exclusion criteria are missing). 
-            “Not all medical charts had adequate documentation regarding laboratory findings such as CSF results; some were only mentioned as “normal”. To avoid the exclusion of these normal findings, a mean of the reference interval was used (2.5 WBC/mm2 for leukocytes and 0.55 g/L for protein in CSF).” – these patients should have been excluded or further search strategies for missing data (e.g. search the laboratory database) should have been applied – this is more important as additional testing on the samples was applied. 
-            A table containing relevant information regarding the included studies in the literature search should be included
-            The heterogeneity of included studies should be discussed 

Minor issues:
-            The statistics should be detailed more. 
-            The first part of the results is hard to understand. You present a pie chart of symptoms in children and adults but the mean age in the next paragraph is between 46-76 years? 
Comments on the Quality of English Language

There are only minor issues detected than can be easily corrected. 

Author Response

Comment 1: The search strategy is not clearly defined. In most review articles more than one database should be searched. The inclusion and exclusion criteria for the studies are not clearly defined. Moreover, the authors fail to describe how studies were selected (how many reviewers screened the articles, how did they manage the cases in which reviewers not in consent etc). Did the authors apply any strategy for missing data (e.g. contact the authors) as just exclusion of the studies is not an academic approach?

Response 1: Thank you for the useful comment. The search strategy has been accordingly clarified in the methods section, including exclusion and inclusion criteria, and how many authors screened the manuscript (line 79-90).

We did not apply any strategy for the reports with missing data other than excluding these.

We chose to only search PubMed for reports and not do a full systematic literature review as this was beyond our scope for the paper. We wanted to present a representable literature cohort to compare with the danish findings, not to present a comprehensive review of every case found in the literature. The recent paper on post-herpetic NMDARE from J. Cleaver et al, BRAIN 2024 (https://doi.org/10.1093/brain/awad419), did a comprehensive review from 2007-2023 including several databases, identifying in total 126 patients. Thus, the 80 patients identified by our literature search using only PubMed from 2009-2021, seems as a representable cohort. 

Comment 2:  It is not clear how patients in the Danish cohort were included (e.g. inclusion and exclusion criteria are missing). 

Response 2: Thank you for bringing this to our attention. Inclusion criteria how now been added to this section (line 128-131).  

Comment 3: “Not all medical charts had adequate documentation regarding laboratory findings such as CSF results; some were only mentioned as “normal”. To avoid the exclusion of these normal findings, a mean of the reference interval was used (2.5 WBC/mm2 for leukocytes and 0.55 g/L for protein in CSF).” – these patients should have been excluded or further search strategies for missing data (e.g. search the laboratory database) should have been applied – this is more important as additional testing on the samples was applied.

Response 3: We appreciate this concern. Unfortunately, we did not have the permission to search laboratory databases for the missing values. We chose not to exclude these patients, as an exclusion would introduce a false high proportion of patients with CSF changes in our HSE cohort.  

Comment 4:  A table containing relevant information regarding the included studies in the literature search should be included

Response 4: Our Supplementary Table 1 includes information reference, sex and age on all the 80 patients included in the literature study. 

Comment 5:   The heterogeneity of included studies should be discussed 

Response 5: Thanks for this observation. The heterogeneity of the studies have now been added as a limitation in the limitations section (line 467-469). 

Comment 6: The statistics should be detailed more. 

Response 6: Thank you for this comment. Is it possible to elaborate on how/what should be more detailed in the description of the statistics? 

Comment 7: The first part of the results is hard to understand. You present a pie chart of symptoms in children and adults but the mean age in the next paragraph is between 46-76 years? 

Response 7: We apologize for the confusion. Figure 2 refers to the next section ("Cases from the literature") and has now been placed accordingly in the text (after Table 1). 

Reviewer 2 Report

Comments and Suggestions for Authors

The authors have prepared a great well-written article addressing the epidemiology of NMDARE in Denmark. This manuscript gives a good impression and may be demanded by clinicians as well as by researchers studying the fundamental aspects of HS infection.

I only have a couple of concerns to consider before the manuscript is published:

-       In Figure 2 the legend has «Abnormal movements» twice. I’d also recommend you indicate the total number of observations

-       Even though the language looks very fine, some sentences need to be modified. Like in « Danish patients were generally treated less aggressive, as fewer received no treatment (only 67% received first line therapy). Additionally, when treated, they were treated less aggressively, as none received second line therapy.» or «adult cohort and less aggressive treatment strategy in Denmark».  «Aggressive» and «aggressively» sound inappropriate here. Also, it’s “second-line therapy”.

-       As I understood based on the information given in Table 2 (Age, years, median (range), in particular), it’s not “a severe lack of children diagnosed with post-HSE NDMARE”, it’s a lack of children diagnosed with HSE in general. If only I hadn’t overlooked something.

-       Optional: Since you noticed the fact of fewer medical interventions in Danish patients, you may try to check the same tendency reflected in previous reports. It’s not even needed to stay within the same nosology if you want to declare the common peculiarities of the Danish healthcare system. Maybe someone has already reported the same tendency in other diseases. And if the last is the case you can speculate in this direction.

Author Response

Comment 1: In Figure 2 the legend has «Abnormal movements» twice. I’d also recommend you indicate the total number of observations

Response 1: Thank you for bringing this to our attention. Figure 2 has been updated and the number of observations has been added to the figure.

Comment 2: Even though the language looks very fine, some sentences need to be modified. Like in « Danish patients were generally treated less aggressive, as fewer received no treatment (only 67% received first line therapy). Additionally, when treated, they were treated less aggressively, as none received second line therapy.» or «adult cohort and less aggressive treatment strategy in Denmark».  «Aggressive» and «aggressively» sound inappropriate here. Also, it’s “second-line therapy”.

Response 2: We appreciate this comment. The sentences have now been modified. Please see line 463-465.

Comment 3: As I understood based on the information given in Table 2 (Age, years, median (range), in particular), it’s not “a severe lack of children diagnosed with post-HSE NDMARE”, it’s a lack of children diagnosed with HSE in general. If only I hadn’t overlooked something.

Response 3: The available HSE database we included in this study, did only contain information on adult cases. Screening a pediatric database would without a doubt be preferable, giving our results, unfortunately one such was not existing. Thus, we cannot draw any conclusion on whether there is a lack of children with HSE in DK.

Comment 4: Optional: Since you noticed the fact of fewer medical interventions in Danish patients, you may try to check the same tendency reflected in previous reports. It’s not even needed to stay within the same nosology if you want to declare the common peculiarities of the Danish healthcare system. Maybe someone has already reported the same tendency in other diseases. And if the last is the case you can speculate in this direction.

Response 4: We appreciate this consideration. When looking at our national AE database, we generally see a timely and appropriate treatment strategy in this cohort of patients. Actually, NMDARE patients in DK are even treated more often with “maintenance therapy” such as oral steroids, azathioprine or mycophenolate mofetil when compared to other cohorts (Reference 25). Based on this. We do not believe that there is reason to suspect a more restrictive treatment approach in DK generally.

Round 2

Reviewer 1 Report

Comments and Suggestions for Authors

The authors addressed our concerns and the manuscript changed accordingly.